# New 20 m Progressive Shuttle Test Protocol and Equation for Predicting the Maximal Oxygen Uptake of Korean Adolescents Aged 13–18 Years

**DOI:** 10.3390/ijerph16132265

**Published:** 2019-06-27

**Authors:** Sang-Hyun Lee, Jung-Ran Song, Yang-Jung Kim, Su-Jin Kim, Hyuk Park, Chang-Sun Kim, Hyo-Bum Kwak, Ju-Hee Kang, Dong-Ho Park

**Affiliations:** 1Department of Kinesiology, Inha University, Incheon 22212, Korea; 2Department of Pharmacology and Medical Toxicology Research Center, Inha University, Dongduck Women’s University, Seoul 02748, Korea; 3Department of Physical Education, Dongduck Women’s University, Incheon 22212, Korea

**Keywords:** VO_2max_, 20 m progressive shuttle test, protocol, adolescent, validity, reliability

## Abstract

*Background*: Although several equations for predicting VO_2max_ in children and adolescents have been reported, the validity of application of these equations to the Korean population has not been verified. The purpose of study was to develop and validate regression models to estimate maximal oxygen uptake (VO_2max_) using a newly developed 20 m progressive shuttle test (20 m PST) protocol in Korean male (*n* = 80, 15.3 ± 1.86 years) and female (*n* = 81, 15.5 ± 1.73 years) adolescents aged 13–18 years. *Methods*: The modified 20 m PST was performed and the VO_2max_ was assessed in a sample of 161 participants. The participants underwent a treadmill test (TT) in the laboratory and the modified 20 m PST in a gymnasium. For the validation study, the participants performed the TT with a stationary metabolic cart and the 20 m PST with a portable metabolic cart once. In addition, they performed the 20 m PST two more times to establish test–retest reliability. *Results*: The mean VO_2max_ (49.6 ± 8.7 mL·kg^−1^·min^−1^) measured with the potable metabolic cart was significantly higher than that measured in the graded exercise test with the stationary metabolic cart (46.6 ± 8.9 mL·kg^−1^·min^−1^, *p* < 0.001) using the new 20 m PST protocol. The standard error of the estimate (SEE) between these two measurements was 1.35 mL·kg^−1^·min^−1^. However, the VO_2max_ derived from the newly developed equation was 46.7 ± 7.3 mL·kg^−1^·min^−1^ (*p* > 0.05) and the SEE was 2.90 mL·kg^−1^·min^−1^. The test and retest trials of the 20 m PST yielded comparable results (laps, *r* = 0.96; last speed, *r* = 0.93). *Conclusions*: Our data suggest that the new 20 m PST protocol is valid and reliable and that the equation developed in this study provides a valid estimate of VO_2max_ in Korean male and female adolescents aged 13–18 years.

## 1. Introduction

In the United States, 24.8% of adolescents perform physical activities on more than five days a week, for 60 min or more per day, whereas only 13.1% of Korean adolescents meet these standards [1]. In addition, globally, 20% of school-going adolescents aged 11–17 years met the recommended guideline of a minimum daily 60 min of moderate-to-vigorous physical activity (PA). The rate of PA in Korean youth is lower than the WHO levels [2]. Lower levels of PA lead to a decline in fitness levels, which is a major cause of obesity in the young population and has become a serious social problem. In developed countries, one measure to solve this social problem is the development and implementation of nationwide youth fitness and health promotion programs (FITNESSGRAM, European Fans in Training, Trim & Fit program, The New Physical Education) [3].

Cardiorespiratory fitness (CRF) is one of the most important fitness factors and is closely related to heart disease, pulmonary disease, type 2 diabetes, and other diseases [4,5]. There is also mounting evidence that CRF is more likely to predict mortality than other established risk factors such as smoking, hypertension, high cholesterol, and type 2 diabetes [6]. Similar to western countries, the 20 m progressive shuttle test (20 m PST) is used to measure CRF levels in Korean adolescents. The graded exercise test (GXT) is generally used to make accurate cardiopulmonary fitness measurements, but it is costly and can only be used in a limited number of individuals because of time constraints [7]. Furthermore, the 20 m PST is used as an alternative to the GXT in elementary, middle, and high schools because of the lack of laboratory equipment required to measure GXT in Korea.

The original 20 m PST developed by Léger et al. [8] is less tedious than long-distance running or step tests and is modified according to the participant’s ability in order to minimize accidents. It has been widely used in schools because it increases student interest and achievement levels by using music and tempo changes [9]. Moreover, the original 20 m PST [8] is the most reliable and valid test related to health [10]. However, the 20 m PST protocol was developed for adolescents from western countries, who have a relatively high fitness level compared with Korean adolescents [11]. Our previous study [12] on second-year middle school female Korean students showed that the initial velocity (8.5 km·h^−1^, increase by 0.5 km·h^−1^) is very high in the existing 20 m PST protocol, resulting in a short test duration (3′59″ ± 1′08″), which may reduce the accuracy of estimated maximal oxygen uptake (VO_2max_). The study also reported a lower correlation (*r* = 0.60) and a small but significant difference between VO_2max_ predicted from the 20 m PST (34.57 ± 3.36 mL·kg^−1^·min^−1^) and VO_2max_ determined on a treadmill (36.89 ± 6.07 mL·kg^−1^·min^−1^) in Korean participants aged 13 years [12]. This is because VO_2max_ is obtained as the product of cardiac output and arteriovenous oxygen difference and suggests that the time taken for the heart to reach maximum cardiac output in the incremental exercise test exceeds 5 min [13,14,15,16]. It was found that excessive speed at initial stages can cause premature interruption of the test and may jeopardize a participant’s health, leading to an under- or overestimation of physical fitness, especially in individuals with low cardiorespiratory fitness levels [17].

Furthermore, validation data for the Korean population are limited [12,18]. In Korea, a personalized physical fitness system (Physical Activity Promotion System, PAPS) was established in 2006 [19]. The 20 m PST has been administered to tens of thousands of children and adolescents as part of a national physical fitness test every year since 2006. Although several equations for predicting VO_2max_ in children and adolescents [20,21,22,23,24,25,26,27,28] have been reported, the validity of these equations in the Korean population has not been verified.

We previously conducted a study where we tested the validity of the 20 m PST in second-year middle school girls aged 14 years [12], and observed that the initial speed of this test (i.e., 8.5 km·h^−1^) was probably excessively demanding in such girls (a short test duration, 3′59″ ± 1′08″). Hence, we adapted the 20 m PST [8] to a modified 20 m PST for assessing CRF in middle school girls aged 13–15 years in a feasible and reliable way [18]. The major adaptation consisted of simply reducing the initial speed from 8.5 km·h^−1^ to 7.5 km·h^−1^. In addition, we observed that the modified 20 m PST is maximal in Korean middle school girls, with an averaged test duration of 5′17″ ± 1′44″ [18]. We also showed that the modified 20 m PST was reliable and valid in girls aged 13–15 years. However, reducing the initial speed (from 8.5 km·h^−1^ to 7.5 km·h^−1^) by 1 km·h^−1^ had a limitation of increasing the test duration by over 5 min. For instance, 45.4% of participants lasted less than 5 min in the modified 20 m PST. In other words, the initial speed of 7.5 km·h^−1^ still seemed to be excessively demanding for Korean girls aged 13 to 15 years.

Therefore, the aims of this study were (1) to provide a modified 20 m PST protocol for 13 to 18 year-old Korean adolescents that can last at least 5 min and (2) to develop a VO_2max_ estimation equation with validity and reliability from the modified 20 m PST protocol.

## 2. Methods

### 2.1. Participants

A convenience sampling method was used to recruit 180 adolescents (90 boys and 90 girls) aged 13–18 (15.4 ± 1.79 years) years from two schools in Incheon (West of Korea). However, ten male and 11 female adolescents were excluded due to discomfort or distress during the test, and technical problems during the test or problems while downloading data, which may have led to inaccurate VO_2max_ results. Thus, a total of 161 adolescents (80 boys and 81 girls, 11% dropout rate) were enrolled in the study. A comprehensive verbal description of the nature and purpose of the study, as well as clinical implications of the investigation was provided to the participants, their parents, and their teachers; this information was also sent to the participants’ parents by surface mail. Written informed consent was obtained from the participants’ parents, in addition to verbal assent from the participants. The inclusion criteria included nonsmokers, no history of cardiovascular or metabolic diseases, no musculoskeletal injuries, nonpregnant status, and no medications during the duration of study. The study protocol was approved by the Institutional Ethics Committee of Inha University, Korea.

### 2.2. Modified 20 m PST Protocol Development

The modified 20 m PST protocol involved 2 min of preparation time at 5.0 km·h^−1^. After completion of the preparatory exercise, the modified protocol was initiated at a speed of 5.0 km·h^−1^ (speed of the 1st preparatory exercise) and was increased by 0.75 km·h^−1^ incrementally. This protocol was designed to maintain an exercise duration of at least 5 min (Table 1).

The modified 20 m PST protocol was developed on the basis of the treadmill incremental load protocol of the Korea Institute of Sports Science (KISS). The GXT protocol used in this study is a modification of the KISS protocol targeting women and men. The modified protocol is initiated at a speed of 5.0 km·h^−1^, with a slope of 3% (for both sexes), over the 2 min warm-up period. Thereafter, the speed is incrementally increased by 1.5 km·h^−1^ every 2 min, whereas the slope is maintained at 3% for both sexes. Meanwhile, the modified 20 m PST protocol is also initiated at a speed of 5.0 km·h^−1^ over the 2-min warm-up period. Thereafter, the speed is incrementally increased by 0.75 km·h^−1^ every 1 min.

### 2.3. Anthropometric Measurements

Height and weight of the participants were measured to the closest 0.1 cm and 0.1 kg, respectively, using a measuring device (TBF-2002; Tanita Co., Tokyo, Japan), without shoes, and wearing light shorts and a t-shirt. The waist-circumference (WC) was measured to the nearest 0.1 cm using a nonelastic tape at the level of the narrowest point between the lower costal border and the iliac crest; hip circumference (HC) was measured at the widest region. Waist–hip ratio (WHR) was calculated as WC (cm) divided by HC (cm). The body fat percentage (% fat) of the participants was measured using an impedance-type body composition analyzer (BIA). BIA (InBody 4.0, Biospace, Seoul, Korea) measurements were undertaken at least two hours after breakfast with an empty bladder. Participants were instructed to refrain from any strenuous exercises 48 h before the test. However, participants were not required to fast before the measurement. The InBody 4.0 body composition analyzer has in-built hand and foot electrodes. Participants wore normal shorts and a t-shirt while standing upright: hands held the electrodes and feet were placed on the electrodes. Age, height, and gender were manually entered after weight was determined by a scale positioned within device. Anthropometric measurements were performed between 8:00 am and 12:00 am. All tests were conducted by the same investigators.

### 2.4. Laboratory Assessment of VO_2max_ (GXT)

VO_2max_ was measured using the KISS GXT protocol on a treadmill after the participants (80 boys, 81 girls) were connected to a wireless heart rate (HR) monitor (POLAR, New York, NY, USA) and were stabilized for 10 min. The initial load of the protocol was set at 5.0 km·h^−1^ (slope: 3%; for both sexes) for 2 min, which was incrementally increased by 1.5 km·h^−1^ every 2 min thereafter. Oxygen uptake (VO_2_, mL·kg^−1^·min^−1^) was measured via open circuit spirometry using an automated gas analyzer (Parvo Medics TrueOne 2400, Sandy, UT, USA), which had been calibrated previously with standard gases. During the tests, gas exchange data were collected continuously using an automated breath-by-breath system. The total time taken to the all-out in each subject was divided with a filtering interval of 20 s, as used in the Parvo Medics software. To ensure that the participants achieved VO_2max_, the measurements were further analyzed when at least two of the following criteria were met: (1) detection of a plateau in the VO_2_ curve, (2) respiratory exchange ratio greater than 1.1, (3) a rate of perceived exertion ≥17 using Borg’s scale, and/or (4) achievement of an aged-predicted maximal heart rate of ±10 bpm (206.9 − [0.67 × age]) [29].

### 2.5. Field Assessment of VO_2max_ (Modified 20 m PST)

The participants wore a portable gas analyzer (K4b^2^, Cosmed, Rome, Italy) to directly measure VO_2max_ during the modified 20 m PST. Before each test, the oxygen and carbon dioxide analyzers were calibrated according to the manufacturer’s instructions. During each test, a gel seal was used to help prevent air leaks from the face mask. Respiratory parameters were recorded breath by breath, which were averaged in turn over a 20 s period. The weight of the K4b^2^ was 1.5 kg, which include the battery and a specially designed harness. In a previous study, McLaughlin and colleagues [30] reported that it is a valid device when compared to the Douglas bag method. Wearing the portable gas analyzer during the 20 m shuttle run test does not significantly alter the participants’ energy demands [31].

The test was conducted according to the modified 20 m PST protocol. In this test, a portable gas analyzer was used to record respiratory parameters every 20 s during testing, while participants inspired room air through a facemask. The maximal oxygen uptake was the main parameter determined using the open circuit method. Before measurement, the gas analyzer was calibrated with standard gases. Exhaustion was confirmed when at least two of the following criteria were met (1) detection of a plateau in the VO_2_ curve, (2) respiratory exchange ratio greater than 1.1, (3) a rate of perceived exertion ≥17 using Borg’s scale, and/or (4) achievement of an aged-predicted maximal heart rate of ±10 bpm (206.9 − [0.67 × age]) [29]. To verify the validity of the modified 20 m PST protocol, the VO_2max_ of the GXT was used.

In the modified 20 m PST, oxygen uptake, HR (Polar RS400, Polar, USA), peak velocity, maximum number of repetitions, final stage velocity, and exercise duration were measured immediately after each exercise phase and at the end of the exercise. VO_2_ measurements with the GXT were performed in a laboratory and VO_2_ measurements with the modified 20 m PST were performed in a gymnasium to ensure similar measurement environments (temperature, 25 °C; humidity, 40%).

All participants performed the modified 20 m PST, as described by Léger et al. [5], on a wooden gymnasium floor. In brief, the participants were required to walk or run between two lines that were 20 m apart, while keeping pace with audio signals emitted from a prerecorded CD. However, unlike the study by Léger et al. [8], the initial velocity was not 8.5 km·h^−1^ but was 5 km·h^−1^, and the velocity was increased by 0.75 km·h^−1^ (not by 0.5 km·h^−1^) each minute (one minute equals one stage). The participants were instructed to run in a straight line, pivot, turn on completing a shuttle, and pace themselves in accordance with the audio signals. The test ended when the participant stopped because of fatigue, or when he/she failed to be within 3 m of the end lines on two consecutive tones. Each subject was encouraged to keep running for as long as possible [32].

### 2.6. Reliability Testing of the Modified 20 m PST Protocol

Test–retest reliability of the modified 20 m PST was ensured in the 161 subjects who participated in the GXT. Measurements were performed twice, at intervals of at least three days, without the use of a portable gas analyzer. The maximum HR (HRmax), final speed (km·h^−1^), number of reps (reps), final stage (stage), and exercise duration (min.s excluding warm-up) were measured in the modified 20 m PST.

### 2.7. Statistical Analysis

Pearson’s correlation coefficients, paired sample *t*-tests, and two-by-two mixed ANOVAs were used to examine the relationship and systemic bias between the VO_2max_ measured with stationary gas analyzer and the VO_2max_ measured with the portable gas analyzer, as well as between the test and retest, to estimate the reliability of the study. A multiple regression analysis was used (*n* = 161) to predict VO_2max_ from the number of laps completed in the modified 20 m PST, sex, age, and weight. The validity of the modified 20 m PST was investigated using Pearson’s correlation coefficients and the 95% LoA method originally reported by Bland and Altman [33] for directly measured VO_2max_ and estimated VO_2max_ from the equation developed in the current study. The Bland–Altman plot for the directly measured VO_2max_ and the estimated VO_2max_ from the equation developed in the current study shows the mean difference (bias) between the measured and estimated VO_2max_ as well as the 95% limits of agreement (mean difference + 1.96 SD of the difference). For all statistical tests, the α level adopted for significance was a two-tailed *p* < 0.05. SPSS version 18.0 (SPSS Inc., Chicago, IL, USA) was used for statistical analyses.

## 3. Results

All participants performed both the GXT and the modified 20 m PST and fulfilled predetermined exhaustion criteria. Descriptive characteristics of the participants are presented as means and standard deviations in Table 2. Excluding the test duration (*p* > 0.05), the modified 20 m PST VO_2max_ (*p* < 0.01), last speed (*p* < 0.01), and HRmax (*p* < 0.01) values obtained using the portable gas analyzer were higher than those measured in the GXT on a treadmill (Table 3). The modified 20 m PST VO_2max_ measured by the portable gas analyzer was 2.95 mL·kg^−1^·min^−1^ (the standard error of estimate; SEE = 1.35 mL·kg^−1^·min^−1^) higher than the VO_2max_ measured in the GXT on a stationary gas analyzer. However, the difference was rather large at a lower speed or stage but decreased with increasing speed or stage (Figure 1). The correlation of the values measured (VO_2max_, last speed, test duration, and HRmax) using the two measurement methods (stationary vs. portable) was statistically significant (Table 3).

A multiple regression analysis was performed to develop an estimation equation to predict VO_2max_ (mL·kg^−1^·min^−1^). In the VO_2max_ estimation equation of the modified 20 m PST, when number of laps (x, laps), age (x, age), sex (x, male = 1, female = 2), and body weight (x, kg) were used as independent variables, it is the best representative of the dependent variable, GXT measured VO_2max_. The modified 20 m PST VO_2max_ estimation formula obtained from the regression analysis is as follow:Y (mL·kg^−1^·min^−1^) = 0.301 × laps − 0.9 × age − 6.642 × G − 0.173 × W + 63.168
with *r* = 0.82; *r*^2^ = 0.67; SEE = 2.90 mL·kg^−1^·min^−1^, *n* = 152.where “laps” are the number of laps in the modified 20 m PST and “age”, “G”, and “W” are the age, sex (boy = 1, girl = 2), and weight, respectively, of the participants performing the modified 20 m PST.

Bland–Altman plots were used to represent random and systematic errors in the directly measured VO_2max_ (Figure 2a) between the GXT and modified 20 m PST, as well as between the directly measured VO_2max_ from GXT and the estimated VO_2max_ from the new equation (Figure 2b). The directly measured VO_2max_ values from the GXT and modified 20 m PST were 46.61 ± 8.92 mL·kg^−1^·min^−1^ and 49.56 ± 8.73 mL·kg^−1^·min^−1^, respectively (*p* < 0.001). The standard error of the estimate (SEE) was 1.35 mL·kg^−1^·min^−1^. The mean difference in VO_2max_ between the GXT and modified 20 m PST was 2.95 mL·kg^−1^·min^−1^ (95% CI). The errors in the measured VO_2max_ from the modified 20 m PST and the correlation coefficients between the estimated and directly measured VO_2max_ are presented in Table 3. However, the directly measured VO_2max_ from the GXT was 46.61 ± 8.92 mL·kg^−1^·min^−1^, with a corresponding mean estimated VO_2max_ of 46.70 ± 7.33 mL·kg^−1^·min^−1^ (*p* > 0.05) from the new equation of this study. The SEE was 2.90 mL·kg^−1^·min^−1^ (*r* = 0.82, *p* < 0.001), which was slightly large but similar to the SEE of VO_2max_ measured from the GXT and modified 20 m PST (Figure 3).

According to the results, no interaction effect was observed for the number of laps, HRmax, final speed, and test duration, excluding the warm-up (Table 4). Male adolescents had significantly higher values than those of female adolescents for all variables. However, all variables except for the number of laps, that is, the maximum speed, test duration, and HRmax, were significantly higher in the first trial than in the second trial. The correlations between the variables measured in the first and second trials were very high (number of laps, *r* = 0.955, *p* < 0.001; last speed, *r* = 0.932, *p* < 0.001; test duration, *r* = 0.944, *p* < 0.001; HRmax, *r* = 0.795, *p* < 0.001). Figure 4 shows the scatterplot of the measured VO_2max_ values and the duration of the modified 20 m PST without the portable gas analyzer. The average duration (min.s) of the 1st and 2nd modified 20 m PST trials was 8.31 ± 1.1.40 (range from 5.10 to 12.19) in male and female adolescents; 9.41 ± 1.21 (range from 6.08 to 12.19) in male adolescents; and 7.24 ± 1.04 (range from 5.10 to 10.21) in female adolescents (Figure 4).

## 4. Discussion

Concerning the first aim of this study, it was observed that: (1) the modified 20 m PST protocol, which involves an increase in the initial speed of 5.0 km·h^−1^ at a rate of 1.5 MET (0.75 km·h^−1^) per minute, seems acceptable for Korean adolescents aged 13–18 years; (2) with respect to the accuracy of VO_2max_ measurement, the time required for the 20 m PST ends within a minimum of 5 min to a maximum of 13 min (12′19″) for both male and female adolescents (Figure 4); (3) the mean difference in VO_2max_ between the GXT and the modified 20 m PST with the use of a gas analyzer was 2.95 mL·kg^−1^·min^−1^ (95% CI).

The original 20 m PST developed by Léger et al. [8] involves an increase in speed by 0.5 km·h^−1^ and an initial speed of 8.5 km·h^−1^. This is a running test method that involves an increase in speed by one metabolic equivalent (MET) per stage. It differs from the modified 20 m PST protocol developed in this study in two aspects: initial speed (5 km·h^−1^) and stage load (0.75 km·h^−1^, 1.5 MET·stage^−1^) (Table 1).

The original 20 m PST protocol, currently used worldwide, was developed by Léger et al. [8]. This protocol was developed on the basis of the VO_2max_ measured from the GXT on a treadmill. The Astrand, Bruce, Balke, Naughton, and KISS protocols are the most widely used protocols of the GXT for VO_2max_ measurement in South Korea. These protocols (2–3 min per stage with an initial speed of 1.6–8 km·h^−1^, inclination of 0–10%, and stage load of 0.9–3.2 METs) vary widely. From this point of view, the protocol developed in this study, which involves an increase in the initial speed of 5.0 km·h^−1^ at a rate of 1.5 MET (0.75 km·h^−1^) per min, seems acceptable for Korean adolescents aged 13–18 years.

The modified 20 m PST developed in this study has the following advantages. First, the initial speed (5.0 km·h^−1^ instead of 8.5 km·h^−1^) is low, which reduces the risk of accidents. Second, with respect to the accuracy of VO_2max_ measurement, the time required for the 20 m PST ends within a minimum of 5 min to a maximum of 13 min (12′19″) for both male and female adolescents (Figure 4). According to the results of this study, the test duration (min.s) of the modified 20 m PST was 8.31 ± 1.40 (range from 5.10 to 12.19) in male and female adolescents; 9.41 ± 1.21 (range from 6.08 to 12.19) in male adolescents, and 7.24 ± 1.04 (range from 5.10 to 10.21) in female adolescents. Previous studies [14,15,16,17] have suggested that high exercise intensity leads to short exercise duration because of the lack of muscle power, whereas a low exercise load leads to a prolonged GXT duration, resulting in a lower VO_2max_ value. Lepretre et al. [34] and McCole et al. [35] reported that the maximum cardiac output was reached in 5–9 min in relation to the cardiovascular response during the GXT in adult male subjects (VO_2max_ = 50.70 mL·kg^−1^·min^−1^). Therefore, there is an optimal incremental exercise test duration to ensure accurate VO_2max_ measurement: at least 5 min and generally 10 ± 2 min [17,34,35,36]. Thus, the modified 20 m PST protocol developed for male and female adolescents in this study is appropriate.

The mean VO_2_ values for stages 1–5 obtained from the GXT and the modified 20 m PST were 32.97 mL·kg^−1^·min^−1^ and 36.41 mL·kg^−1^·min^−1^, respectively. The VO_2_ values obtained from the modified 20 m PST were about 10.4% higher than those obtained from the GXT. The difference between the VO_2max_ from the GXT (46.61 ± 8.92 mL·kg^−1^·min^−1^) and from the modified 20 m PST (49.56 ± 8.73 mL·kg^−1^·min^−1^) was 6.3%, and the VO_2max_ value obtained from the modified 20 m PST was high (Figure 1). Thus, directly measured VO_2max_ from the modified 20 m PST is significantly (*p* < 0.001) overestimated compared with the directly measured VO_2max_ from the GXT. The mean difference in VO_2max_ between the GXT and the modified 20 m PST was 2.95 mL·kg^−1^·min^−1^ (95% CI). In this study, the modified 20 m PST measurement was considered slightly error-prone as it was performed in the gymnasium on a wooden floor rather than on a treadmill. However, the gym’s flat surface requires a similar energy expenditure to that on a treadmill at 1–2% slope [37] because of the additional energy required to move the body forward and the air resistance when running. In addition, the modified 20 m PST VO_2_ measurement is similar to that obtained from the GXT on a treadmill at 2–3% slope, taking into account the additional energy consumption associated with wearing the VO_2_ measurement equipment (1.5 kg) during the modified 20 m PST. Considering these factors, the VO_2max_ measured from the modified 20 m PST was significantly higher than the VO_2max_ measured from the GXT, although it should be similar to the VO_2max_ measured from the GXT. Flouris et al. [38] measured VO_2max_ using a similar protocol (start speed, 8.5 km·h^−1^) in a laboratory (treadmill) and in the field (the original 20 m PST) using a portable gas analyzer. In the present study, VO_2max_ (46.61 ± 8.92 mL·kg^−1^·min^−1^) measured in the GXT on a treadmill was statistically significantly lower than VO_2max_ (49.56 ± 8.73 mL·kg^−1^·min^−1^) measured in the modified 20 m PST (2.95 mL·kg^−1^·min^−1^; 6.3% difference, *p* > 0.001), similar to the results of a previous study [38]. This suggests that the increase in anaerobic metabolism and biomechanical factors induced in the 20 m PST, unlike that in the GXT, increases VO_2_ [18,26,39,40,41]. In other words, the VO_2_ increases because of additional energy consumption due to vertical movement of the human body’s center of gravity during the 20 m PST and increase in anaerobic metabolism due to the repeated acceleration and deceleration motions at the start and finish lines. Another assumption is that the mean VO_2_ difference at stages 1–5 between the two measurements (GXT vs. 20 m PST) was greater than the VO_2max_ difference measured from the two measurements because the GXT on the treadmill is easy to control through treadmill speed, whereas the 20 m PST tries to match the sound source, but the subject tends to move faster than the scheduled rate because he/she must walk or run by predicting the speed. This leads to additional energy consumption, and this difference is reduced by reaching the final stage of increasing speed, which is believed to be because of a decrease in the difference in VO_2max_ between the measurements (GXT vs. 20 m PST).

Regarding the second aim of this study, it was found that: (1) the directly measured VO_2max_ from the GXT was 46.61 ± 8.92 mL·kg^−1^·min^−1^, with a corresponding mean estimated VO_2max_ of 46.70 ± 7.33 mL·kg^−1^·min^−1^ (*p* > 0.05) from the new equation of this study. the SEE was 2.90 mL·kg^−1^·min^−1^, which was slightly high but similar to the SEE (1.35 mL·kg^−1^·min^−1^) of VO_2max_ measured from the GXT and the modified 20 m PST. the correlation coefficient (*r* = 0.82; *r*^2^= 0.67) between the estimated and directly measured VO_2max_ was high; (2) on testing the reliability of the modified 20 m PST, the number of laps (*r* = 0.96), final speed (*r* = 0.93), test duration (*r* = 0.94), and HRmax (*r* = 0.80) were found to be highly correlated in the 1st and 2nd repeated trials; (3) VO_2max_ can be indirectly estimated from the modified 20 m PST in Korean adolescents aged 13–18 years using the new equation.

To test the validity of the modified 20 m PST developed in this study, Bland–Altman plots were used to represent the random and systematic errors between the directly measured VO_2max_ from GXT and the estimated VO_2max_ from the new equation (Figure 2b). The directly measured VO_2max_ from the GXT was 46.61 ± 8.92 mL·kg^−1^·min^−1^, with a corresponding mean estimated VO_2max_ of 46.70 ± 7.33 mL·kg^−1^·min^−1^ (*p* > 0.05) from the new equation of this study. The SEE was 2.90 mL·kg^−1^·min^−1^, which was slightly high but similar to the SEE (1.35 mL·kg^−1^·min^−1^) of VO_2max_ measured from the GXT and the modified 20 m PST. The correlation coefficient (*r* = 0.82; *r*^2^ = 0.67, *p* < 0.001) between the estimated and directly measured VO_2max_ was high. According to Léger & Lambert [42], Léger et al. [8], and Park et al. [18], the correlation coefficients (r) between the VO_2max_ predicted from the 20 m PST and VO_2max_ measured from the GXT on a treadmill were *r* = 0.84, *r* = 0.89, and *r* = 0.74, respectively. Flouris et al. [38] also showed that when the VO_2max_ was measured using the same protocol (starting speed, 8.5 km·h^−1^) in a laboratory (treadmill) and in the field (20 m PST) using a portable gas analyzer, the correlation coefficient (r) was 0.86–0.91. These results show that the 20 m PST protocol developed in this study is suitable for male and female adolescents because similar correlation coefficients (*r* = 0.82, *p* < 0.001) were found. Also, the SEE value was small (2.90 mL·kg^−1^·min^−1^), indicating the validity of the newly developed equation. Ruiz et al. [28] reported that the SEE ranged from 5.30 mL·kg^−1^·min^−1^ [27] to 6.50 mL·kg^−1^·min^−1^ [8]. In addition, recent studies have reported that the SEE ranges from 1.25 mL·kg^−1^·min^−1^ [25] to 6.53 mL·kg^−1^·min^−1^ [43].

On testing the reliability of the modified 20 m PST developed in this study, the number of laps (*r* = 0.96, *p* < 0.001), final speed (*r* = 0.93, *p* < 0.001), test duration (*r* = 0.94, *p* < 0.001), and HRmax (*r* = 0.80, *p* < 0.001) were found to be highly correlated in the 1st and 2nd repeated trials. Whereas there was no significant difference in the number of laps, final speed, or test duration between the 1st and 2nd trials, HRmax was significantly lower in the 2nd trial than in the 1st trial. The number of laps, HRmax, final speed, and test duration were significantly greater for male adolescents (Table 4). According to Lamb & Rogers [44], in the second trial, the number of laps may be increased compared to that in the first trial due to the learning effect, but there was no statistically significant difference between the 1st and 2nd trials. These results differ from those of a previous study [35]. The reason for this is that the participants in this study were already familiar with the 20 m PST and thus their results may be slightly different from the results of the participants (i.e., 35 male college students) in the study by Lamb & Rogers [44].

This study has several limitations that should be taken into account in future studies. First, the equation to estimate VO_2max_ in modified 20 m PST was obtained from a sample of Korean adolescents aged 13 to 18 years, so the validity of the equation should be tested in other populations and by laboratory methods. Second, participants of this study live in one specific area of Korea and the sample size is small. Therefore, there is a limit to the generalization of the results of this study. On the other hand, this study has several strengths and weaknesses. The original 20 m PST [8] has been known to be the most reliable and valid test related to health [10]. However, excessive speed at the initial stage (i.e., 8.5 km·h^−1^) may cause premature interruption of the exercise test, consequently leading to the under- or overestimation of CRF, especially in individuals with low CRF levels. This is the first study to provide a protocol that allows adolescents with low fitness levels to continue testing for at least 5 min to ensure accuracy of CRF testing, while adolescents with high fitness levels can end the test within a maximum of 13 min (12′19″). This is also the first study that provides an equation that allows indirectly estimation of VO_2max_ in Korean adolescents aged 13–18 years. A weakness of this study is that the measurement time of the modified 20 m PST is longer than the existing original protocol [8]. Although further research is needed to confirm the results of this study, the modified 20 m PST proposed seems to be a useful tool for measuring CRF in Korean adolescents aged 13–18 years.

## 5. Conclusions

The modified 20 m PST protocol developed in this study has the following advantages. First, the initial speed (5.0 km·h^−1^ instead of 8.5 km·h^−1^) is low, which may reduce the risk of accidents. Second, with respect to the measurement accuracy of VO_2max_, the duration required for a modified 20 m PST seems to be appropriate (male = 9.87 ± 1.35 min and female = 7.60 ± 1.06 min).

The VO_2max_ predicted by the prediction equation developed using the modified 20 m PST protocol was highly correlated with the VO_2max_ measured by the GXT (*r* = 0.82). Also, the SEE value was acceptable (2.90 mL·kg^−1^·min^−1^). This seems to indicate the validity of the modified 20 m PST protocol. In the reliability test of the modified 20 m PST, the number of laps (*r* = 0.96), last speed (*r* = 0.93), test duration (*r* = 0.94), and HRmax (*r* = 0.80) were highly correlated and had high reproducibility on repeated measurements. Therefore, the modified 20 m PST protocol developed in this study and the prediction equation for estimating VO_2max_ are appropriate for Korean male and female adolescents. However, additional studies are needed to determine whether this modified 20 m PST protocol and prediction equation are applicable to adults and other age groups (elementary school students, seniors, etc.).

## Figures and Tables

**Figure 1 ijerph-16-02265-f001:**
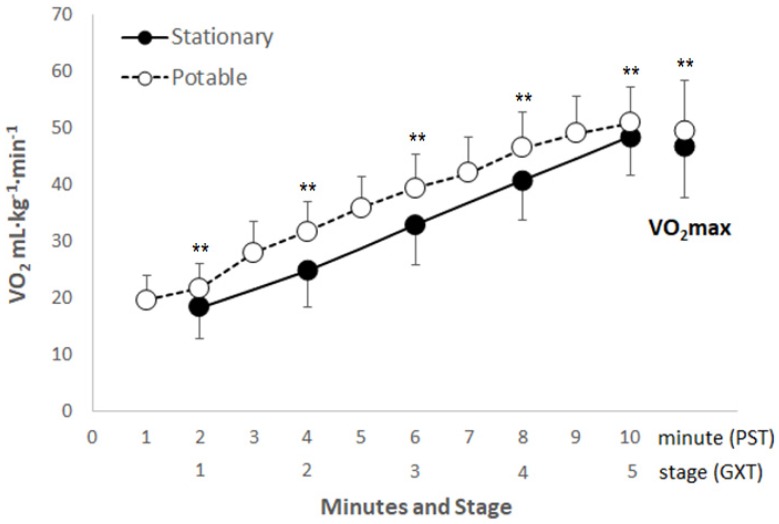
Scattergram of the measured oxygen uptake (VO_2_) at each minute (or stage) and the maximal VO_2_ in male and female adolescents. Asterisks (**) indicate statistically significant (*p* < 0.01) differences compared with the measured VO_2_ from the stationary metabolic cart according to repeated-measures ANOVA.

**Figure 2 ijerph-16-02265-f002:**
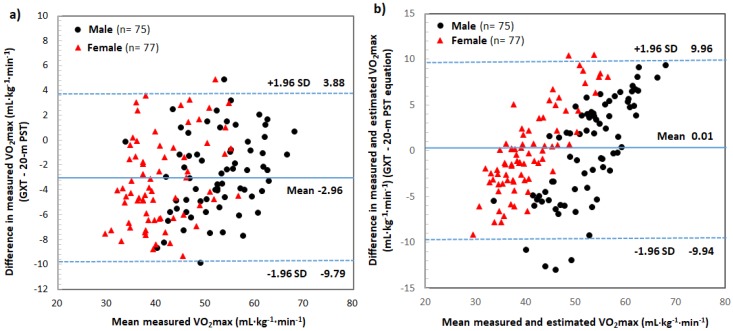
(**a**) Bland–Altman plot for the directly measured VO_2max_ from the GXT and from the modified 20 m PST. (**b**) Bland–Altman plot for the directly measured VO_2max_ and estimated VO_2max_ from the new equation. The solid line represents the mean difference (bias) in the VO_2max_ measured from the GXT and that measured from the modified 20 m PST. The upper and lower broken lines represent the 95% limits of agreement (mean difference ± 1.96 SD of the difference).

**Figure 3 ijerph-16-02265-f003:**
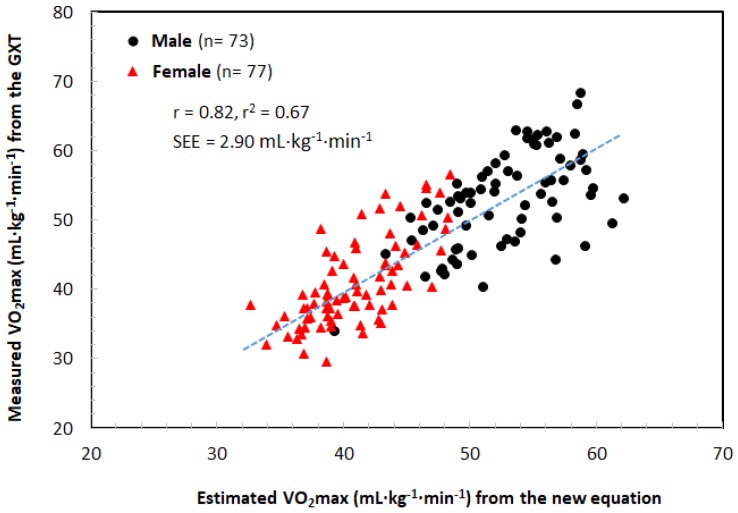
Scatterplot of the measured VO_2max_ from the GXT and the estimated VO_2max_ from the modified 20 m PST.

**Figure 4 ijerph-16-02265-f004:**
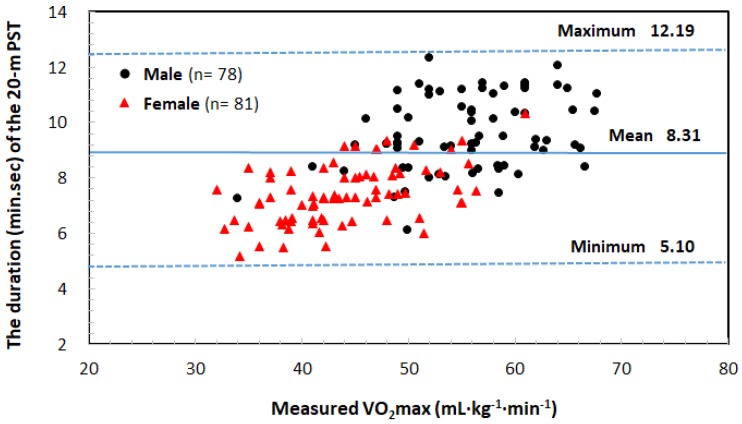
Scatterplot of the measured VO_2max_ and the duration (minutes.seconds) of the modified 20 m PST. The average test duration was 8.31 ± 1.40 (range from 5.10 to 12.19) in male and female adolescents; 9.41 ± 1.21 (range from 6.08 to 12.19) in male adolescents; and 7.24 ± 1.04 (range from 5.10 to 10.21) in female adolescents. The solid line represents the mean duration of the modified 20 m PST. Upper and lower broken lines represent the maximum and minimum durations of the modified 20 m PST.

**Table 1 ijerph-16-02265-t001:** The graded exercise test (GXT) and modified 20 m progressive shuttle test (PST) protocols.

GXT Protocol (Modified KISS Protocol)			Modified 20 m PST Protocol *					
Stage (2 min, 3% Grade Stage)	Speed (km·h^−1^)	Speed (m·min^−1^)	Stage (1 min)	Speed (km·h^−1^)	Lap.	Cumulative Count	20 m Reaching Time (Sec)	Expended Time·Stage ^−1^ (Sec)
2-min warm-up	5.00	83.33		5.00	4		14.40	57.60
	5.00	83.33		5.00	4		14.40	57.60
1	5.00	83.33	1	5.00	4		14.40	57.60
			2	5.75	5	9	12.50	62.60
2	6.50	108.33	3	5.75	5	14	12.50	62.60
			4	7.25	6	20	9.90	59.60
3	8.00	133.33	5	7.25	6	26	9.90	59.60
			6	8.75	7	33	8.20	57.60
4	9.50	158.33	7	9.50	8	41	7.60	60.60
			8	9.50	8	49	7.60	60.60
5	11.00	183.33	9	11.00	9	58	6.50	58.90
			10	11.75	10	68	6.10	61.30
6	12.50	208.33	11	11.75	10	78	6.10	61.30
			12	13.25	11	89	5.40	59.80
7	14.00	233.33	13	14.00	12	101	5.10	61.70
			14	14.00	12	113	5.10	61.70
8	15.50	258.33	15	15.50	13	126	4.60	60.40
			16	15.50	13	139	4.60	60.40
9	17.00	283.33	17	17.00	14	153	4.20	59.30
			18	17.75	15	168	4.10	60.80
10	18.50	308.33	19	17.75	15	183	4.10	60.80
			20	19.25	16	199	3.70	59.80
11	20.00	333.33	21	20.00	17	216	3.60	61.20
			22	20.00	17	233	3.60	61.20
12	21.50	358.33	23	21.25	18	251	3.40	61.00
			24	21.25	18	269	3.40	61.00

Note: The original 20 m PST protocol involves an increase in the speed at a rate of 0.5 mL·kg^−1^·min^−1^, starting with an initial velocity of 8.5 km·h^−1^, without warm up. The modified 20 m PST protocol is initiated at 5.0 km·h^−1^, and the speed is then increased at a rate of 0.75 mL·kg^−1^·min^−1^, but the same intensity is maintained for 2 min by adjusting the number of laps per minute. In addition, the 2-min warm-up and initial speed were equivalent to a total warm-up time of 3 min. For this measurement, a sound source CD for the modified 20 m PST was produced and used.

**Table 2 ijerph-16-02265-t002:** Descriptive characteristics (mean ± SD) of the participants in this study.

Variable	Boys (*n* = 80)	Girls (*n* = 81)	Total (*N* = 161)
Age (year)	15.30 ± 1.86	15.50 ± 1.73	15.40 ± 1.79
Height (cm)	170.60 ± 6.20 *	159.40 ± 5.4	164.99 ± 8.10
Weight (kg)	63.66 ± 9.95 *	54.69 ± 8.59	59.14 ± 10.29
BMI (kg·m^−^^2^)	21.82 ± 3.03	21.49 ± 3.04	21.65 ± 3.03
Body fat (%)	16.10 ± 8.00 *	25.40 ± 8.20	20.77 ± 9.32
Waist-hip ratio	0.79 ± 0.05 *	0.75 ± 0.05	0.77 ± 0.05
VO_2max_ (mL·kg^−1^·min^−1^) ^#^	52.79 ± 6.80 *	40.91 ± 6.33	46.77 ± 8.85

* *p* < 0.001, compared with girls. ^#^ According to valid data, *n* for VO_2max_ is lower due to missing data (boys, *n* = 75; girls, *n* = 77). Abbreviations: BMI = body mass index; VO_2max_ = maximal oxygen consumption.

**Table 3 ijerph-16-02265-t003:** Comparison of measured VO_2max_ between stationary and potable gas analyzers.

Variable	*n*	Stationary	Portable	SEE	*p*	r ^#^
VO_2max_ (mL·kg^−1^·min^−1^)	140	46.61 ± 8.92	49.56 ± 8.73 **	1.350	0.000	0.922 **
Last speed (km·h^−1^)	140	10.60 ± 1.65	10.95 ± 1.31 **	0.787	0.001	0.724 **
Test duration (min. sec)	142	8.10 ± 2.24	7.58 ± 1.47	1.262	0.169	0.689 **
HRmax (bpm)	142	195.74 ± 7.51	199.01 ± 6.82 **	5.991	0.000	0.578 **

** *p* < 0.01, differences against the measured VO_2_ using a stationary metabolic cart with paired *t*-test. ^#^ Pearson’s correlation (r) between the stationary and portable values. Abbreviations: SEE = the standard error of the estimate.

**Table 4 ijerph-16-02265-t004:** Reliability analysis (test–retest) results of the modified 20 m PST in boys and girls.

Variable	Trial	Male (*n* = 78)	Female (*n* = 81)	Total (*n* = 159)	*p*	*r* ^#^
# of Laps (repetition)	1st	70.43 ± 14.64 ^a,^**	48.48 ± 9.18	59.25 ± 16.38	T: 0.276 S: 0.000 T × S: 0.817	0.955 **
	2nd	69.62 ± 13.11 ^a,^**	48.15 ± 10.07	58.83 ± 15.95		
Last speed (km·h^−1^)	1st	12.24 ± 1.12 ^a,^**	10.49 ± 9.18	11.35 ± 1.30	T: 0.447 S: 0.000 T × S: 0.447	0.932 **
	2nd	12.18 ± 1.00 ^a,^**	10.49 ± 10.07	11.32 ± 1.26		
Test duration (min.sec)	1st	9.40 ± 1.28 ^a,^**	7.25 ± 1:04	8.32 ± 1.42	T: 0.628 S: 0.000 T × S: 0.475	0.944 **
	2nd	9.41 ± 1.19 ^a,^**	7.22 ± 1:11	8.30 ± 1.42		
HRmax (bpm)	1st	204.24 ± 7.90 ^a,^*^,^^b,^**	201.35 ± 7.05 ^b,^**	202.77 ± 7.59	T: 0.000 S: 0.008 T × S: 0.809	0.795 **
	2nd	201.92 ± 7.53 ^a,^**	198.84 ± 7.05	200.35 ± 7.43		

* *p* < 0.05, ** *p* < 0.01, two-way repeated ANOVA with Bonferroni post hoc significance level. ^a^ Significantly different from the values in girls, ^b^ significantly different from the values in the 2nd trial. ^#^ Pearson’s correlation (r) between trials for the whole sample. Abbreviations: T = trial; S = sex; HRmax = maximum heart rate.

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
