# Peer review of "New 20 m Progressive Shuttle Test Protocol and Equation for Predicting the Maximal Oxygen Uptake of Korean Adolescents Aged 13–18 Years"

_ijerph, 2019, doi:10.3390/ijerph16132265_

Round 1

Reviewer 1 Report

Comments in pdf. 

Author Response

First of all, thank you for your scrupulous and kind comments. Our manuscript has been much improved owing to your suggestions. Thank you so much!

 Specific comments

Page 1 Lines 11: please specify number of males and females, mean age and standard deviation

Response: Thank you for your comments. We added the number of males and females, mean age, and standard deviation to the data. 

Page 1 Lines 29: I suggest to give examples of some more countries.

Response: Thank you for your comments. We added WHO data on Prevalence of Insufficient Physical Activity Data.

Page 2 Lines 42: please correct the numbers of sections in the whole manuscript. there are several mistakes

Response: We apologize for the typos. We have corrected the numbers of sections in the manuscript; 1à 2

Page 2 Lines 44: add mean and standard deviation

Response: Thank you for your comments. We have added mean age and standard deviation data. 

Page 2 Lines 45: please explain the reasons with more details. This is important.

Response: We described the reasons for exclusion in detail in page 2, lines 49-50: “technical problems during the test or problems while downloading data, which may have led to inaccurate VO2max results.

Page 4 Lines 12: please specify the instrument used to measure waist and hip. Was Waist–hip ratio calculated?

Response: We described the instrument in more detail in lines 12-15: The waist-circumference (WC) was measured to the nearest 0.1 cm using a nonelastic tape at the level of the narrowest point between the lower costal border and the iliac crest; hip circumference (HC) was measured at the widest region. Waist-hip ratio (WHR) was calculated as WC (cm) divided by HC (cm).”

Page 4 Lines 13: you have to give many more details here about the procedure. at what time? how? were all the recommendations for an adequate impedance analysis followed? please describe

Response: We described the instrument in more detail in lines 15-24; Height and weight of the participants were measured to the closest 0.1 cm and 0.1 kg, respectively, using a measuring device (TBF-2002; Tanita Co., Japan), without shoes, and wearing light shorts and a t-shirt. The waist-circumference (WC) was measured to the nearest 0.1 cm using a nonelastic tape at the level of the narrowest point between the lower costal border and the iliac crest; hip circumference (HC) was measured at the widest region. Waist-hip ratio (WHR) was calculated as WC (cm) divided by HC (cm). The body fat percentage (% fat) of the participants was measured using an impedance-type body composition analyzer (BIA). BIA (InBody 4.0, Biospace, Inc. Korea) measurements were undertaken at least two hours after breakfast with an empty bladder. Participants were instructed to refrain from any strenuous exercises 48 h before the test. However, participants were not required to fast before the measurement. The InBody 4.0 (Biospace, Inc. Korea) body composition analyzer has in-built hands and feet electrodes. Participants wore normal shorts and a t-shirt while standing upright: hands held the electrodes and feet were placed on the electrodes. Age, height, and gender were manually entered after weight was determined by a scale positioned within device. Anthropometric measurements were performed between 8:00 am and 12:00 am. All tests were conducted by the same investigators.”

Page 5 Lines 49: please correct the number

Response: We apologize for the typos. We have corrected them: 2à 3

Page 10 Lines 7: please correct number

Response: We apologize for the typos. We have corrected them: 3à 4

Page 12 Lines 10: please correct number

Response: We apologize for the typos. We have corrected them: 4à 5

Page 12 Lines 9: Please add a final paragraph in the discussion with the limittaions and srengths of the study. & Page 12 Lines 22: please state what should be the design of these future studies.

Response:

Response: We have added a section on the strengths, the weaknesses, and the limitations of the study, and design of future research in the end of the discussion as followings: This study has several limitations that should be taken into account in future studies. First, the equation to estimate VO2max in modified 20-m PST was obtained from a sample of Korean adolescents aged 13 to 18 years, so the validity of the equation should be tested in other populations and by laboratory methods. Second, participants of this study live in one specific area of Korea and the sample size is small. Therefore, there is a limit to the generalization of the results of this study. On the other hand, this study has several strengths and weaknesses. The 20-m PST (Léger et al. 1988) has been known to be the most reliable and valid test related to health (Ruiz et al. 2011). However, excessive speed at the initial stage (i.e., 8.5 km·h-1) may cause premature interruption of the exercise test, consequently leading to the under- or over-estimation of CRF, especially in individuals with low CRF levels. This is the first study to provide a protocol that allows adolescents with low fitness levels to continue testing for at least 5 min to ensure accuracy of CRF testing, while adolescents with high fitness levels can end the test within a maximum of 13 min. This is also the first study that provides an equation that allows indirectly estimation of VO2max in Korean adolescents aged 13-18 years. A weakness of this study is that the measurement time of the modified 20-m PST is longer than the existing original protocol (Léger et al. 1988). Although further researches are needed to confirm the results of this study, the modified 20-m PST proposed seems to be a useful tool for measuring CRF in Korean adolescents aged 13-18 years.”

Reviewer 2 Report

Park et al. attempted to develop a validated regression equation of the 20m progressive shuttle run test for estimating cardiorespiratory fitness of Korean adolescents. Overall, it is extremely difficult to follow and understand the manuscript due to poor English. In addition, the manuscript has a number of outstanding issues to be addressed;

(Introduction)

-       Overall, Introduction should be improved to support the rationale of the study to modify the original 20-m PST for Korean adolescents. Especially, the authors need to provide a better explanation for the reason(s) how the initial speed of the 20-m PST by Léger et al. limits the accuracy of the estimation regression. What are the validity and reliability when applied to Korean adolescents?

-       In addition, fitness difference among children in Western and Korea as one of the reasons for developing a new regression equation of the 20-m PST should be supported with review of literatures.

(Methods)

-       Section of participants needs to be better described in a detailed manner. Are they representative of Korean adolescents?

-       With respect to the modified 20-m PST protocol, the authors need to clarify the starting speed of the 20-m PST protocol; 8.5 km×h-1 in the first sentence vs. 5.0 km×h-1 later? Why did the authors decide 8.5 or 5.0 km×h-1 as a starting speed?

-       Explain the KISS GXT protocol in a detailed manner.

-       Consider to remove Table 1 (redundant in text).

-       What is the purpose of comparing VO2max values measured with GXT and PST? 

-       Explain why PST-based VO2max is higher than GXT-based one?

-       The reliability and validity of estimated VO2max values should be compared between the original 20-m PST developed by Léger et al and the modified one by the authors et al.

-       In addition, a cross-validation of the modified PST should be conducted by applying it to another sample of Korean adolescents.

(Discussion)

-       Discussion should be improved in its context and English also. In the current format, it is difficult to follow.

-       Lines (18-19): ‘From this point of view, the protocol developed in this study, which involves an increase in the initial speed of 5.0 km·h-1 at a rate of 1.5 MET (0.75 km·h-1) per minute, seems acceptable for the general public.’ I do not agree with this statement because the modified protocol has not applied to the general population.

-       (Line 22): ‘which reduces the risk of accidents’. Explain the risk of accidents.

-       (Lines 22-35): ‘Second, with 22 respect to the accuracy of VO2max measurement, the duration of the 20-m PST was at least 5 min 23 but less than 13 min for both male and female adolescents.’ It is extremely difficult to follow. Please rewrite the sentences.

-       Overall, difficult to follow due to poor English. Consider overall rewriting.

-       Study limitations should be mentioned.

Author Response

First of all, thank you for your kind comments. The quality of our manuscript has been improved due to your suggestions. We owe it to you.

Comments and Suggestions for Authors

Park et al. attempted to develop a validated regression equation of the 20m progressive shuttle run test for estimating cardiorespiratory fitness of Korean adolescents. Overall, it is extremely difficult to follow and understand the manuscript due to poor English. In addition, the manuscript has a number of outstanding issues to be addressed;

Response: I apologize for any inconvenience caused by the language. English is a second language for me and my manuscript has been checked by a professional English editing service. I thank you for your understanding.

(Introduction)

- Overall, Introduction should be improved to support the rationale of the study to modify the original 20-m PST for Korean adolescents. Especially, the authors need to provide a better explanation for the reason(s) how the initial speed of the 20-m PST by Léger et al. limits the accuracy of the estimation regression. What are the validity and reliability when applied to Korean adolescents?

Response: As pointed out, I added the rationale of the study to modify the original 20-m PST for Korean adolescents as follows (page 2, lines 10-22): “Moreover, the original 20-m PST [8] is the most reliable and valid test related to health [10]. However, the 20-m PST protocol was developed for adolescents from western countries, who have a relatively high fitness level compared with Korean adolescents [11]. Our previous study [12] on second-year middle school female Korean students showed that the initial velocity (8.5 km·h-1, increase by 0.5 km·h-1) is very high in the existing 20-m PST protocol, resulting in a short test duration (3'59” ± 1'08”), which may reduce the accuracy of estimated maximal oxygen uptake (VO2max). The study also reported a lower correlation (r = .60) and a small but significant difference between VO2max predicted from the 20-m PST (34.57 ± 3.36 mL·kg-1·min-1) and VO2max determined on a treadmill (36.89± 6.07 mL·kg-1·min-1) in Korean participants aged 13 years [12].

- In addition, fitness difference among children in Western and Korea as one of the reasons for developing a new regression equation of the 20-m PST should be supported with review of literatures.

Response: As pointed out, I added the reference as below:

Gill, J.M.; Celis-Morales, C.A.; Ghouri, N. Physical activity, ethnicity and cardio-metabolic health: does one size fit all? Atherosclerosis. 2014, 232(2), 319-33.

(Methods)

- Section of participants needs to be better described in a detailed manner. Are they representative of Korean adolescents?

Response: As pointed out, the participants are not representative of Korean adolescents. So, we added the implications for future research in the end of the discussion as followings (in page 11, lines 31-32); “Participants of this study live in one specific area of Korea and the sample size is small. Therefore, there is a limit to the generalization of the results of this study.”

- With respect to the modified 20-m PST protocol, the authors need to clarify the starting speed of the 20-m PST protocol; 8.5 km×h-1 in the first sentence vs. 5.0 km×h-1 later? Why did the authors decide 8.5 or 5.0 km×h-1 as a starting speed?

Response: I apologize for the confusion. It was written incorrectly. We removed 8.5 mL·kg-1·min-1 in parentheses and corrected the sentence in page 3 lines 13.

- Explain the KISS GXT protocol in a detailed manner.

Response: we address the KISS GXT protocol in page 4 lines 1-6 as follows: “The GXT protocol used in this study is a modification of the KISS protocol targeting women and men. The modified protocol is initiated at a speed of 5.0 km·h-1, with a slope of 3% (for both sexes), over the 2-min warm-up period. Thereafter, the speed is incrementally increased by 1.5 km·h-1 every 2 min, whereas the slope is maintained at 3% for both sexes.”

-  Consider to remove Table 1 (redundant in text).

Response: Although I have considered deleting it as you mentioned, it explains the test protocol. I hope you understand.

- What is the purpose of comparing VO2max values measured with GXT and PST? 

Response: We wanted to check the validity of the measurement between the two. According to a previous research, an estimation equation using a portable gas analyzer was also developed. Therefore, we wanted to compare the difference when using a stationary gas analyzer called a gold standard and whether the difference is due to the modified 20-m PST protocol or due to the characteristics of the 20m PST. In addition, if there were any differences, we wanted to know the components. It was concluded that this difference was due to the characteristics of the 20m PST: 1) continuous run (stationary) vs. intermittent run & repetitive turn (portable) 2) running according to the speed of the treadmill (scheduled) vs. speed control by yourself (not constant, sometimes faster than the set speed).

  -  Explain why PST-based VO2max is higher than GXT-based one?Response: This is described in the discussion section as follows (page 10 line 29-39): “In other words, the VO2 increases because of additional energy consumption due to vertical movement of the human body’s center of gravity during the 20-m PST and increase in anaerobic metabolism due to the repeated acceleration and deceleration motions at the start and finish lines. Another assumption is that the mean VO2 difference at stages 1–5 between the two measurements (GXT vs. 20-m PST) was greater than the VO2max difference measured from the two measurements because the GXT on the treadmill is easy to control through treadmill speed, whereas the 20-m PST tries to match the sound source, but the subject tends to move faster than the scheduled rate because he/she must walk or run by predicting the speed. This leads to additional energy consumption, and this difference is reduced by reaching the final stage of increasing speed, which is believed to be because of a decrease in the difference in VO2max between the measurements (GXT vs. 20-m PST).

- The reliability and validity of estimated VO2max values should be compared between the original 20-m PST developed by Léger et al and the modified one by the authors et al.

Response: We totally agree with the reviewer’s suggestion. There is no direct comparison to estimated VO2max because of different protocols (Léger et al. initial speed = 8.5 km·h-1 vs. the authors et al. initial speed = 5 km·h-1). Thus, the standard error of the estimate (SEE) and correlation coefficients (r) values were compared between these two measurements in page 11 line 2-15.

- In addition, a cross-validation of the modified PST should be conducted by applying it to another sample of Korean adolescents.

Response: We totally agree with the reviewer’s suggestion. However, additional experiments are required to meet your suggestions. Therefore, the problem you mentioned was described in the section on research limitations in page 11 line 28-33.

(Discussion)

- Discussion should be improved in its context and English also. In the current format, it is difficult to follow.

Response: I apologize for any inconvenience caused by the language. English is a second language for me and my manuscript has been checked by a professional English editing service. I thank you for your patience.

- Lines (18-19): ‘From this point of view, the protocol developed in this study, which involves an increase in the initial speed of 5.0 km·h-1 at a rate of 1.5 MET (0.75 km·h-1) per minute, seems acceptable for the general public.’ I do not agree with this statement because the modified protocol has not applied to the general population.

Response: We agree to your suggestion. So, we modified the sentence as follows (page 9, lines 25-27): From this point of view, the protocol developed in this study, which involves an increase in the initial speed of 5.0 km·h-1 at a rate of 1.5 MET (0.75 km·h-1) per min, seems acceptable for Korean adolescents aged 13-18 years.

- (Line 22): ‘which reduces the risk of accidents’. Explain the risk of accidents.

Response: There was a reason for planning this study. In Korea, the 20m PST is used to evaluate CRF at the elementary, middle, and high school levels quarterly. Unfortunately, a few years ago in Korea, a boy in middle school was in a state of brain death due to an accident during the 20m PST. The reason was that the student's fitness level was extremely low and the initial speed of the test (i.e., 8.5 km·h-1) was probably excessively demanding for the boy. This led to a conclusion that it was necessary to develop safer a 20-m PST protocol. In other words, the idea was to develop a protocol that included warm-ups and lower initial rates (i.e., 5 km·h-1), and if possible, to develop maximal and sub-maximal VO2max estimation equations for CRF. So we analyzed the respiratory gas during the modified 20-m PST as well as laboratory measurements. 

- (Lines 22-35): ‘Second, with 22 respect to the accuracy of VO2max measurement, the duration of the 20-m PST was at least 5 min 23 but less than 13 min for both male and female adolescents.’ It is extremely difficult to follow. Please rewrite the sentences.

Response: We apologize for the confusion. We corrected it as follows: “the duration of the 20-m PST was at least 5 min 23 but less than 13 min for both male and female adolescents à the time required for the 20 m PST ends within a minimum of 5 min to a maximum of 13 min (12’19”) for both male and female adolescents (Figure 4). (page 9, lines 30-31).

- Overall, difficult to follow due to poor English. Consider overall rewriting.

Response: I apologize for any inconvenience caused by the language. English is a second language for me and my manuscript has been checked by a professional English editing service. I thank you for your patience.

- Study limitations should be mentioned.

Response: As pointed out, we added the limitations at the end of the discussion in page 11, line 28-32 as followings; This study has several limitations that should be taken into account in future studies. First, the equation to estimate VO2max in modified 20-m PST was obtained from a sample of Korean adolescents aged 13 to 18 years, so the validity of the equation should be tested in other populations and by laboratory methods. Second, participants of this study live in one specific area of Korea and the sample size is small. Therefore, there is a limit to the generalization of the results of this study.

Reviewer 3 Report

ijerph-509806 New 20-m Progressive Shuttle Test Protocol and Equation for Predicting the Maximal Oxygen Uptake of Korean Adolescents Aged 13–18 Years

General comments

Use the term VO2max consistently throughout the manuscript in the same manner

Use min instead of minutes throughout the manuscript

Use in the tables consistently two or one decimal, the same in the text, e.g. for VO2max-values

Specific comments

Page 1 Lines 36-37: add a reference

Page 2 Lines 1-3: add a reference

Page 2 Lines 7-9: add a reference

Page 2 Line 20: tens of thousands must be adapted to the journal style (Palatino Linotype)

Page 2 Line 41: what is your hypothesis?

Page 2 Line 43: how were the subjects recruited? What were the criteria for exclusion to the study?

Page 10 Line 7: I suggest starting the discussion with (a) what was the intention of the study and (b) what were the main findings. Make then a structure in the discussion with your main findings comparing to results of the literature

Page 11 Line 39: Lambert

Page 12 Line 10: please add the strength, the weakness, the limitations and the implications for future research

Author Response

First of all, we thank you for your generous and kind comments. Our manuscript is improved due to your suggestions. We owe it to you. Thank you so much!

General comments

 Use the term VO2max consistently throughout the manuscript in the same manner

Response: We apologize for the typos. We corrected the typos throughout the manuscript;

 Use min instead of minutes throughout the manuscript

Response: We have corrected it throughout the manuscript;

 Use in the tables consistently two or one decimal, the same in the text, e.g. for VO2max-values

Response: We have corrected it throughout the manuscript. We consistently use two decimal in the tables and the text.

 Specific comments

 Page 1 Lines 36-37: add a reference

Response: As pointed out, I added a reference as below:

Leite, S.A.; Monk, A.M.; Upham, P.A.; Bergenstal, R.M. Low cardiorespiratory fitness in people at risk for type 2 diabetes: early marker for insulin resistance. Diabetol. Metab. Syndr. 2009, 1(1), 8.

Steell, L.; Ho, F.K.; Sillars, A.; Petermann-Rocha, F.; Li, H.; Lyall, D.M.; Iliodromiti, S.; Welsh P, Anderson, J.; et al. Dose-response associations of cardiorespiratory fitness with all-cause mortality and incidence and mortality of cancer and cardiovascular and respiratory diseases: the UK Biobank cohort study. Br J Sports Med. 2019, doi: 10.1136/bjsports-2018-099093.

 Page 2 Lines 1-3: add a reference

Response: As pointed out, this could be a situation in Korea, I added “in Korea” to the sentence instead of a reference.

 Page 2 Lines 7-9: add a reference

Response: As pointed out, I added a reference as below:

Gill, J.M.; Celis-Morales, C.A.; Ghouri, N. Physical activity, ethnicity and cardio-metabolic health: does one size fit all? Atherosclerosis. 2014, 232(2), 319-33.

 Page 2 Line 20: tens of thousands must be adapted to the journal style (Palatino Linotype)

Response: As pointed out, we modified it to a journal style (Palatino Linotype).

 Page 2 Line 41: what is your hypothesis?

Response: I agree that it is a fundamental principle to establish a hypothesis and prove it in most studies. However, this study is not a research study with a comparative or control group or proves a research hypothesis through direct comparison of existing and modified protocols. This study attempted to solve the problem of accuracy of the measurements that can be caused by the high initial velocity for adolescents with low fitness levels, as suggested in preliminary study. Second, it simply describes a new protocol and equation with reliability and validity. For this reason, it describes the research purpose instead of the research hypothesis. However, as pointed out, the purpose of this study seemed to be unclear and I have rewritten clearly as follows (page 2, lines 41-43): Therefore, the purpose of this study was to develop a new VO2max estimation formula with reliability and validity through the development of a modified 20-m PST protocol that allows the male and female adolescents aged 13 to 18 years to last at least 5 min by increasing the increment of the stage speed (from 1 MET·stage-1  to 1.5 MET·stage-1) instead of lowering the initial speed (5.0 km·h-1).àTherefore, the aims of this study were 1) to provide a modified 20-m PST protocol for 13 to 18 years-old Korean adolescents that can last at least 5 min and 2) to develop a VO2max estimation equation with validity and reliability from the modified 20-m PST protocol.”

 Page 2 Line 43: how were the subjects recruited? What were the criteria for exclusion to the study?

Response: We described more in detail in page 2, lines 47-48; A convenience sampling method was used to recruit 180 adolescents (90 boys and 90 girls) aged 13–18 (15.4 ± 1.79 years) years from two schools in Incheon (West of Korea).” And the criteria for inclusion to the study were also described Page 3, Lines 6-7; The inclusion criteria included non-smokers, no history of cardiovascular or metabolic diseases, no musculoskeletal injuries, non-pregnant status, and no medications during the duration of study..” Exclusion criteria are not described separately because they are contrary to the inclusion criteria. Thank you for your understanding.

 Page 10 Line 7: I suggest starting the discussion with (a) what was the intention of the study and (b) what were the main findings. Make then a structure in the discussion with your main findings comparing to results of the literature.

Response: As pointed out, we began the discussion with the main findings in page 9 line 9 – 15 as follows: “Concerning the first aim of this study, it was observed that: (1) the modified 20-m PST protocol, which involves an increase in the initial speed of 5.0 km·h-1 at a rate of 1.5 MET (0.75 km·h-1) per minute, seems acceptable for Korean adolescents aged 13-18 years; (2) with respect to the accuracy of VO2max measurement, the time required for the 20 m PST ends within a minimum of 5 min to a maximum of 13 min (12’19”) for both male and female adolescents (Figure 4); (3) the mean difference in VO2max between the GXT and the modified 20-m PST with the use of a gas analyzer was 2.95 mL·kg-1·min-1 (95% CI).

And also page 10, line 40 – 49 as follows: “Regarding the second aim of this study, it was found that: (1) the directly measured VO2max from the GXT was 46.61 ± 8.92 mL·kg-1·min-1, with a corresponding mean estimated VO2max of 46.70 ± 7.33 mL·kg-1·min-1 (p > 0.05) from the new equation of this study. the SEE was 2.90 mL·kg-1·min-1, which was slightly high but similar to the SEE (1.35 mL·kg-1·min-1) of VO2max measured from the GXT and the modified 20-m PST. the correlation coefficient (r = 0.82; r2= 0.67) between the estimated and directly measured VO2max was high; (2) on testing the reliability of the modified 20-m PST, the number of laps (r = 0.96), final speed (r = 0.93), test duration (r = 0.94), and HRmax (r = 0.80) were found to be highly correlated in the 1st and 2nd repeated trials; (3) VO2max can be indirectly estimated from the modified 20-m PST in Korean adolescents aged 13–18 years using the new equation.”

 Page 11 Line 39: Lambert

Response: We apologize for the typos. We corrected the typos; lambert à Lambert

 Page 12 Line 10: please add the strength, the weakness, the limitations and the implications for future research

Response: As pointed out, we added the strengths, weaknesses, limitations, and implications for future research in the end of the discussion as followings (page 11, lines 28-44); This study has several limitations that should be taken into account in future studies. First, the equation to estimate VO2max in modified 20-m PST was obtained from a sample of Korean adolescents aged 13 to 18 years, so the validity of the equation should be tested in other populations and by laboratory methods. Second, participants of this study live in one specific area of Korea and the sample size is small. Therefore, there is a limit to the generalization of the results of this study. On the other hand, this study has several strengths and weaknesses. The original 20-m PST (Léger et al. 1988) has been known to be the most reliable and valid test related to health (Ruiz et al. 2011). However, excessive speed at the initial stage (i.e., 8.5 km·h-1) may cause premature interruption of the exercise test, consequently leading to the under- or over-estimation of CRF, especially in individuals with low CRF levels. This is the first study to provide a protocol that allows adolescents with low fitness levels to continue testing for at least 5 min to ensure accuracy of CRF testing, while adolescents with high fitness levels can end the test within a maximum of 13 min (12’19”). This is also the first study that provides an equation that allows indirectly estimation of VO2max in Korean adolescents aged 13-18 years. A weakness of this study is that the measurement time of the modified 20-m PST is longer than the existing original protocol (Léger et al. 1988). Although further researches are needed to confirm the results of this study, the modified 20-m PST proposed seems to be a useful tool for measuring CRF in Korean adolescents aged 13-18 years.

Round 2

Reviewer 2 Report

(Majors)

Again, I don’t think that the authors did address the issues related to the validity and reliability of the original 2-m PST. First, Gill et al.’s study has nothing to do with the validity and reliability of the 20-m PST. In that study, they suggested that the optimal level of physical activity and/or fitness required to achieve low cardio-metabolic disease risk may differ according to ethnic group. Second, study sample does not represent boys and girls aged 13-15 in Korea. The proposed regression equation was not cross-validated in another population. So I don’t think that the proposed regression equation for estimating VO2max can be generalized in Korean boys and girls aged 13-15 years, as they stated in Abstract and Introduction.  

(Minors)

The abbreviations (i.e., FITNESSGRAM, 36 EUROFIT, Trim & Fit program, The New PE) should be stated in full names. English still needs to be improved throughout the manuscript by a professional editor.

Author Response

First of all, we thank you for your excellent comments and suggestions. Our manuscript is improved due to your comments and suggestions. Thank you again!

Comments and Suggestions for Authors

 (Majors)

Again, I don’t think that the authors did address the issues related to the validity and reliability of the original 2-m PST. First, Gill et al.’s study has nothing to do with the validity and reliability of the 20-m PST. In that study, they suggested that the optimal level of physical activity and/or fitness required to achieve low cardio-metabolic disease risk may differ according to ethnic group.

Response: Last time, as pointed out “fitness difference among children in Western and Korea as one of the reasons for developing a new regression equation of the 20-m PST should be supported with review of literatures.So, I added the reference as below: Gill, J.M.; Celis-Morales, C.A.; Ghouri, N. Physical activity, ethnicity and cardio-metabolic health: does one size fit all? Atherosclerosis. 2014, 232(2), 319-33. In this review, Gill et al. reported relationship between cardiorespiratory fitness (VO2max) and objectively-measured physical activity in South Asian and European men. Of course, since adults, not youth, participated in that study, it can not be concluded that Asian youths are lower than Western youths. However, we might easily assume that lower levels of PA lead to a decline in fitness levels. In addition, every two years, the National Survey on Physical Fitness is conducted at the national level in Korea (2015 the National Survey on Physical Fitness in Korea; 2017 the National Survey on Physical Fitness in Korea). Since all of them are published in Korean, there are limitations in quoting them. The figure below shows the difference in 20m PST between age and sex of Japanese (red bar) and Koreans (blue bar). Even in comparison with the Japanese, it can be seen that the CRF of Koreans is lower than that of Japanese. Anyway, we corrected a reference (Ghouri et al. 2013) instead of a paper by Gill et al. (2014) as follow:

Ghouri, N.; Purves, D.; McConnachie, A.; Wilson, J.; Gill, J.M.; Sattar, N. Lower cardiorespiratory fitness contributes to increased insulin resistance and fasting glycaemia in middle-aged South Asian compared with European men living in the UK. Diabetologia. 2013,56(10), 2238-49. 

Second, study sample does not represent boys and girls aged 13-15 in Korea. The proposed regression equation was not cross-validated in another population. So I don’t think that the proposed regression equation for estimating VO2max can be generalized in Korean boys and girls aged 13-15 years, as they stated in Abstract and Introduction. 

Response: Thank you for your comments and I agree with your comments. The participants are not representative of Korean adolescents. So, we added the limitation in the end of the discussion (in page 11, lines 31-32). In Abstract and Introduction, what we talk about is validity, not generalization in this study. Therefore, we present the results of Bland-Altman plot and SEE related to the validity. Thank you for your understanding.

 (Minors)

The abbreviations (i.e., FITNESSGRAM, EUROFIT, Trim & Fit program, The New PE) should be stated in full names. English still needs to be improved throughout the manuscript by a professional editor.

Response: Thank you for your comments. Full name is described as follows: EUROFIT = European Fans in Training. The New PE = The New Physical Education. I apologize for any inconvenience caused by the language. As pointed out, our manuscript has been checked by a professional English editing service as follow: 

I thank you for your patience.

Round 3

Reviewer 2 Report

Again, I am not satisfied with the authors' explanations to the following two comments made by myself.

First, the authors speculate that adolescents from Western countries would have higher fitness levels than their counterparts from Korea, based on previous findings comparing Western vs. Asian adolescents or Korean vs. Japanese ones. I don't know how the authors are so sure that that would be the case in Korea?

Second, generalisability or generalization is defined as the extent to which the findings of a study can be applicable to other settings. It is also known as  external validity. Thus, generalization requires internal validity as well as a judgement on whether the findings of a study are applicable to a particular group (called external validity).  Consequently, I suspect that not conducting a cross-validation study would lead to the lack of the external validity.